# Thickness, Adhesion and Microscopic Analysis of the Surface Structure of Single-Layer and Multi-Layer Metakaolin-Based Geopolymer Coatings

Martin Jaskevic [1,*], Jan Novotny [1], Filip Mamon [1], Jakub Mares [1] and Angelos Markopoulos [2]

1   Faculty of Mechanical Engineering, J. E. Purkyne University in Usti nad Labem, Pasteurova 3334/7, 40001 Usti nad Labem, Czech Republic; jan.novotny@ujep.cz (J.N.); mamon@iic.cas.cz (F.M.); mares@iic.cas.cz (J.M.)
2   School of Mechanical Engineering, National Technical University of Athens, Heroon Polytechniou 9, 15780 Athens, Greece; amark@mail.ntua.gr
*   Correspondence: martin.jaskevic@ujep.cz

**Abstract:** This work is focused on creating coating layers made of a metakaolin-based geopolymer suspensions (GP)-formed Al matrix modified using $H_3PO_4$ acid with $Al(OH)_3$ in isopropyl alcohol, named GP suspension I, and $H_3PO_4$ acid with nano $Al_2O_3$ in isopropyl alcohol, named GP suspension J. The selected GP suspensions were applied on aluminum and steel underlying substrates as single-layer coatings and multi-layer coatings, where multi-layer coatings included three and five layers that were polymerized by a curing process. Curing was divided into two types with every layer curing process and final layer curing process. For both GP suspensions I and J, the effect of the number of layers and the type of substrate on adhesion was investigated. The prepared samples on underlying substrates were characterized on the microscopy analysis including SEM for high-resolution images and 3D laser confocal microscopy (CLSM) for the 3D visualization of the coatings structure. Microscopy analysis showed structural defects such as porosity, cracks and peeling, which increase with a greater number of applied layers. However, these defects were only evident on a micro scale and did not seem to be fatal for the performance of the surface stability. The EDS mapping of the prepared layer showed inhomogeneity in the distribution of elements caused by the brush application. A grid test and thickness measurement were performed to complete the microscopy analysis. The grid test confirmed a very high adhesion of GP coatings on the aluminum substrate with a rating of one (only in one case was there a rating of two) and a lower adhesion on the steel substrate with the most frequent rating of three (in one case, there were ratings of two and one). The thickness measurement proved a noticeably thicker thickness of the prepared layer on the Fe substrate compared to the Al substrate by 20%–30% in the case of suspension I and by 70%–10% in the case of suspension J. The thickness of the layer also showed a dependence on the method of application and curing, as a thicker layer was always achieved when curing after the final layer of the GP suspension, compared to curing after each applied layer. The resulting single-layer and multi-layer thicknesses ranged from approx. 7 to 30 μm for suspension I and from approx. 3 to 11 μm for suspension J. A non-linear increase in thickness was also evident from the thickness measurement data.

**Keywords:** geopolymers; single-layer coating; multi-layer coatings; aluminum; construction steel; microstructure characterization; layers thickness; grid test

## 1. Introduction

Synthetic inorganic polymers, also known as geopolymers (GP), consist of chains or networks of mineral molecules linked by covalent bonds [1].

GP are considered to be environmentally friendly materials that have the potential to be used as substitutes for ordinary Portland cement [2,3]. Geopolymers are produced from natural sources such as kaolinite or clays. Modern approaches also try to use industrial

sources including fly ash, waste paper sludge or granulated blast furnace slag. Recycling these resources can have positive environmental impacts and reduce $CO_2$ production compared to Portland cement production [4].

GP, as a group of alkaline-activated materials, have a number of exceptional properties such as strength, resistance to acids and bases, fire resistance, good thermal stability and good adhesion to the underlying substrate [5].

Also, there are some possibilities to use GP as a coating material for metal and non-metal substrates. GP coating properties rely on the chemical composition of raw materials, followed by the roughness of the substrate or the Si/Al ratio [6,7]. Different types and concentrations of the acid could change the behavior and properties of the final GP coatings. For example, Shamala et. al. prepared GP coatings on wood substrates with various NaOH concentrations to find the best concentration for GP coatings [8]. The next factor for GP coatings depends on the water content, which affects the results of the GP coating thickness [9].

The preparation of geopolymers includes three basic phases. Dissolution is the first phase, in which Si and Al atoms transition from the basic raw material to the solution and complexes with hydroxide ions are formed. In the second phase, the condensation of monomers with mobile precursors follows, with a partial internal restructuring of the alkaline polysilicates. The third stage of the geopolymer formation process is the poly-condensation or the polymerization of monomers. Here, the polymer structure is formed, and the whole system solidifies. The product is an inorganic polymer structure. It is very complicated to analyze ongoing processes because they occur almost simultaneously [10].

The geopolymerization process is based on the reaction of reactive aluminosilicates supplemented with metakaolin or fly ash, which quickly dissolve in alkaline solutions in the presence of alkali hydroxides (NaOH/KOH). This creates tetrahedral units-connected polymeric precursors ($-SiO_4-AlO_4-$ or $-SiO_4-AlO_4-SiO_4-$ or $-SiO_4-AlO_4-SiO_4-SiO_4-$) forming amorphous geopolymer products with a 3D network structure [11,12]. Nergis reported that geopolymers also contain three types of pores formed by the arrangement of the OH– and Si groups (Si–OH), Si–O–Si groups, Si–O–Al groups and Si–O rings [13].

Some geopolymers are activated by acidic activators. The metakaolin-based geopolymer produced by using a phosphoric acid solution as an activator has a high compressive strength up to 93.8 MPa [14,15]. Another study also showed that acid-based geopolymers have a higher temperature resistance (up to 1450 °C) and better mechanical properties than alkali-based geopolymers [16]. However, GP have extremely good thermal stability and adhesion to the surfaces thought to be a kind of all-purpose material and potential tribological material [17].

Fire resistance is an interesting property of GP if the suspension contains a flame retardant. In our study, the stability of coatings with $Al(OH)_3$ is monitored for possible future fire protection applications on Fe and Al substrates. $Al(OH)_3$ as a flame retardant in a GP suspension was described on polystyrene and chipboard underlying substrates, and its positive influence has been proven [18].

The adhesion of the coating to the underlying substrate is an essential factor which is controlled by the surface treatments of the base metal as an underlying substrate [18]. Metal with a high surface roughness will have a higher adhesion strength with GP coating material compared to the polished metal substrate [9].

A general theory covering all relevant properties and parameters involved in the design and application of tribological coating composites is very far from being realized. Such a theory would have to treat the long chain of relations ranging from the coating deposition parameters to the tribological response of the coated component [19,20].

Obviously, a good adhesion to the substrate is a crucial property of most applications of coated components. Any adhesion test must superimpose an external stress field over the coating/substrate interface to cause a measurable adhesive failure. Since this stress field will depend on the geometry and type of loading (indentation, scratching, sliding, abrasion, impact, etc.) as well as on the elastic and plastic parameters of the coating and

substrate, the resulting adhesion value will only be representative of the particular test from which it has been obtained [21].

There are many ways in which suspensions can be applied to a substrate surface. One of the simple methods of application, together with the satisfactory results of the final layer, is the application of the suspension with a brush. This method is very cheap, with minimal economic costs in creating a coating, and does not require deep knowledge and know-how in applying and creating a coating. The disadvantage is that it is not possible to accurately correct the achieved layer thickness, and also, the homogeneity of the resulting layer thickness fluctuates within a certain range. An airbrush appears to be another suitable method for applying the suspension. This method is also widely used, but its use is already economically and technologically more demanding. However, it can improve the homogeneity of the thickness of the resulting layer [22].

The aim of the work was to prepare and compare single- and multi-layer coatings on metal underlying substrates, aluminum and construction steel. It was important to explain what effect applying a thicker layer with a brush would have on the adhesion, overall surface quality and change in thickness compared to single-layer systems and various application/curing methods. The prepared GP coatings were characterized using confocal microscopy and SEM (scanning electron microscopy) to capture the microstructure and the visual quality of the coatings. These results were supplemented by a grid test and thickness measurement.

This work follows our previous research [23,24] and expands our knowledge about the selected single- and multi-layer GP coatings with different applications by brushes and some types of curing processes. We observed changes in the visual quality of the prepared GP coatings and changes in their mechanical properties.

## 2. Materials and Methods

The GP coating was selected based on our previous research [23,24]. The selected GP had good properties, which was a good starting point for the following complex research creating multi-layer GP coatings. In this work, we presented this advanced preparation of GP coatings compared with a multi-layered GP coating.

The chosen substrates were aluminum alloy EN-AW 6060 (AlMgSi$_{0.5}$) [25,26] and construction steel 1.0038 (according to EN 10025-2). These underlying substrates have been chosen as the most common alloys in all sectors of industry. The preparation of the substrate before the application of the geopolymer suspension consists only in removing gross impurities and degreasing the surface with an organic solvent (acetone). No other pre-treatment of the surface was applied; therefore, their natural oxide layers are found on the surface of the substrates.

### 2.1. Preparation of the Suspensions and Underlying Substrate

The preparation of the geopolymer suspension consists in mixing basic raw materials that have different phases (liquid and solid). In this research, there is a liquid component, phosphoric acid ($H_3PO_4$) and isopropylalkohol (iPrOH), for both GP suspensions and a solid component, metakaolin with $AlOH_3$ (for GP suspension I) and metakaolin with powder $Al_2O_3$ (for GP suspension J). The good homogenization of the resulting mixture after mixing the basic ingredients is very important. A laboratory homogenizer, AD300L-H, 10,000 RPM, was used for homogenization and mixing.

Geopolymer suspensions I and J were selected from previous research for their interesting properties on the Al substrate [23], where geopolymer I reached an average coating thickness of 2.7 μm and geopolymer J reached one of only 1.5 μm. Both suspensions had excellent adhesion to the Al substrate. Even the microhardness values of HV 0.1 achieved very good results (GP I 118.4 HV 0.1 and GP J 127.1 HV 0.1) compared to the underlying substrate Al 93.6 HV 0.1.

### 2.2. Roughness of Al Substrate EN AW-6060 and Fe Substrate 1.0038

For the application of conventional coatings with an organic or inorganic composition (most often for the anti-corrosion protection of metals) on metal surfaces, the roughness of the underlying substrate is very important [27,28]. For example, for aluminum alloys, the adhesion of such coatings is generally lower than that when applied to steel surfaces [29–33].

The roughness of the used metal substrates EN-AW 6060 and 1.0038 was measured using a Hommel Tester t1000 according to ISO 4287. The input values of the measurement were as follows: probe type T1E 2 μm/90°, compressive force 1.5 mN, traverse length 4.8 mm, traverse speed 0.5 mm/s and measurement range ±80 μm/0.01 μm. For the underlying metal substrates, a sheet with a thickness of 3 mm was used for both types. These sheets were processed by rolling and show a one-way orientation of the grooves that were created during the rolling process, as shown in the detail of the surface in Figure 1. This orientation is very well observed in the aluminum alloy. The roughness of the substrates was measured along the rolling direction A and also perpendicular to the rolling direction B, as shown in Figure 1. Table 1 shows the achieved surface roughness values in individual directions [23].

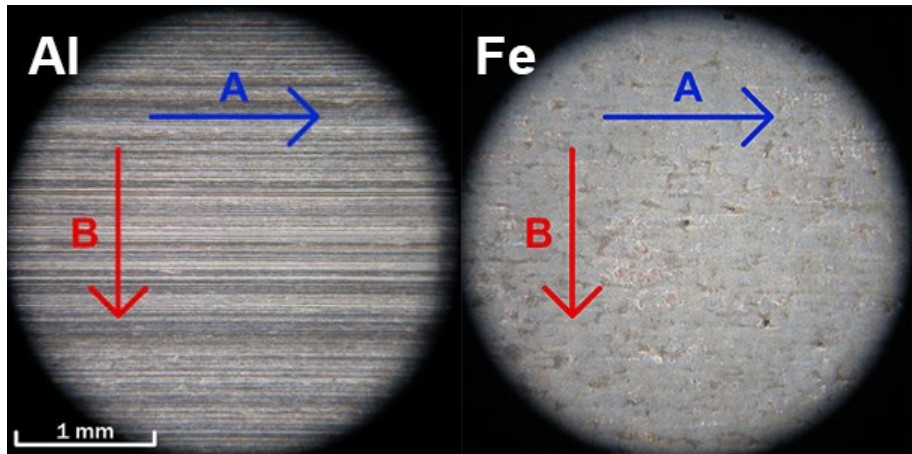

**Figure 1.** Surface detail of the EN-AW 6060 (Al) [23] and 1.0038 (Fe) underlying substrate with visible directional anisotropy and roughness measurement in the direction of rolling A and the perpendicular direction of rolling B.

**Table 1.** Surface roughness of the underlying substrate EN-AW 6060 and 1.0038.

| Substrate | Al | | Fe | |
|---|---|---|---|---|
| | **Measurement Direction** | | **Measurement Direction** | |
| | **A** | **B** | **A** | **B** |
| $R_a$ [μm] | 0.206 | 0.841 | 1.091 | 0.874 |
| $R_z$ [μm] | 1.117 | 5.410 | 6.225 | 5.098 |
| $R_{max}$ [μm] | 1.488 | 6.710 | 6.850 | 6.613 |
| $R_t$ [μm] | 1.603 | 6.958 | 8.448 | 7.538 |

$R_a$—arithmetic mean roughness; $R_z$—ten-point mean roughness; $R_{max}$—maximum roughness depth; $R_t$—maximum height of the profile.

### 2.3. Application of GP Suspensions

Application by brush was chosen, which is the simplest possible application of geopolymers with a sufficient resulting coating quality [23].

The geopolymer suspension was applied to the substrate by a brush, which is designed for water-based coatings. This method was chosen as in previous research [23], but with a different approach regarding the thickness of the coatings. In this case, we tried to prepare

thicker layers with a brush compared to the previous research [23]. In previous research, it was found that by applying geopolymer suspensions with different compositions in one layer using a brush, very thin coatings with a thickness of up to approx. 20 μm (depending on the type of GP) with good adhesion can be achieved on the Al and Fe substrate. These thickness sizes were conditioned by the application of a very thin layer of the suspension with a brush; when applying the suspension, care must be taken to spread it very well over the surface of the substrate. The result was coatings that have a very good surface quality and very good adhesion [23,24]. Such application of GP in a thin layer is not complicated, but it requires concentration, and when applying it to larger or more fragmented surfaces, this procedure may no longer be followed exactly. A failure to follow the procedure can be caused by, for example, the human factor or even the brush application method itself, when this method is simple but not very accurate. The following research therefore simulates a process where the application procedure of a very thin layer is not followed, but the GP layer applied with a brush is thicker and a comparison is made of the effect on the properties (adhesion) and the appearance of the surface that the application of a thicker layer/layers will have.

Labeling explanations in Figure 2: GP—geopolymer suspension, X—type of geopolymer suspension (I or J), -/S—application and curing method (—-curing after each applicated layer; S—curing after three or five layers; see Figure 3), 1 L, 3 L, 5 L—number of layers applied (1 L—one layer, 3 L—three layers, 5 L—five layers). Our geopolymers were divided into four series, where two different GP coatings (I or J) were applied by brush on two metal substrates (Al or Fe).

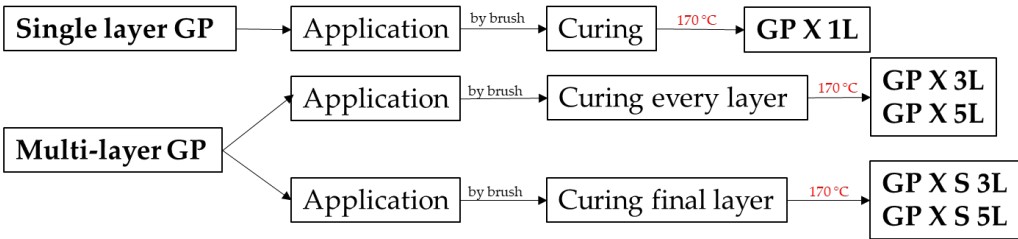

**Figure 2.** Scheme of the preparation of each GP coating.

In order for geopolymer suspensions to acquire their final properties after the application to the underlying substrate (Figure 2), chemical reactions, so-called geopolymerization, must occur in the mixture [24,34,35]. For the geopolymer mixture, geopolymerization occurs at elevated temperatures, in contrast to mixtures with a different composition, where geopolymerization can occur at lower temperatures [24,34,35]. For the selected geopolymer mixture, it is necessary to reach a certain minimum temperature, which was experimentally determined to be 170 °C, and to maintain the same conditions as in the previous research [23]. This temperature is an important parameter influencing the resulting quality of the coatings. This increase in the temperature is needed for the geopolymerization process [24].

Every series of GP has a different curing for the geopolymerization of these coatings shown in Figure 3. The geopolymers marked GP X 1L, GP X 3L and GP X 5L had a curing process with every layer immediately after the application. The GP marked GP X S 3L and GP X S 5L had a different approach to geopolymerization curing, with only single curing after the third or fifth layer of GP coatings. Between every layer, there were 24 h of drying, and after that, the next layer was applied. This approach changed the coatings' properties and their behavior (see the next chapters). This curing procedure was chosen in order to simplify the application of the suspensions to the substrate and at the same time reduce the resulting cost of creating multilayer coatings and analyze whether there is a difference between the layers when applying and curing after each layer compared to curing after applying the final layer.

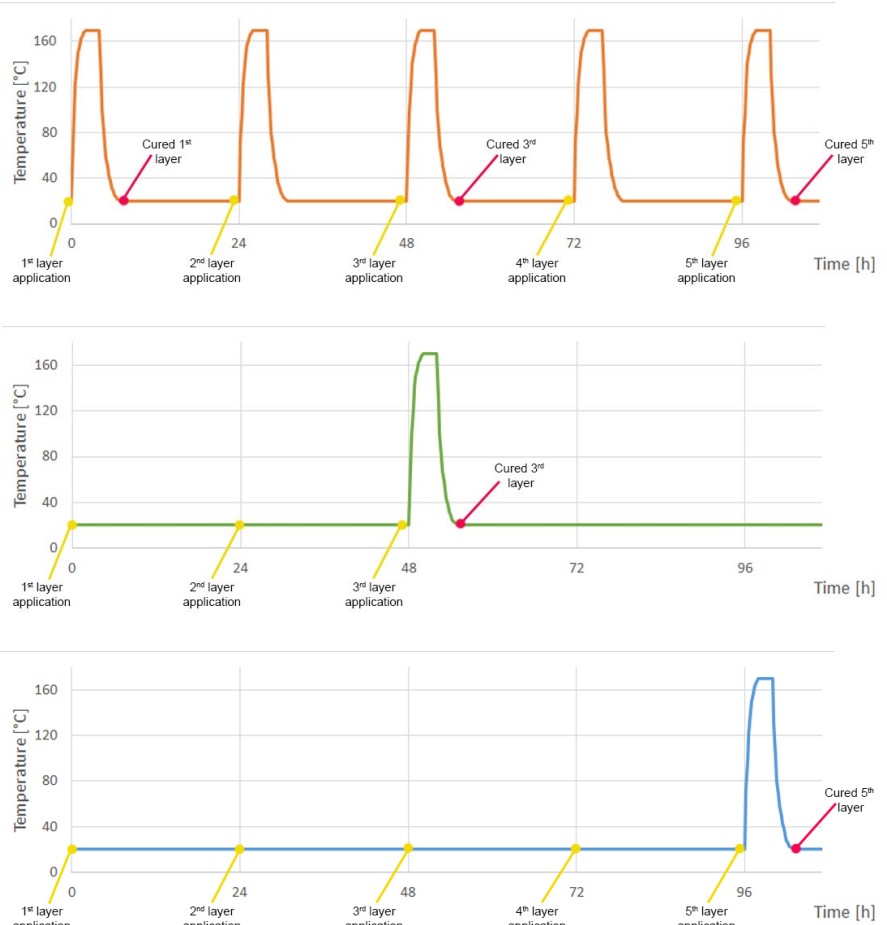

**Figure 3.** Curing for the geopolymerization of coatings.

### 2.4. Experimental Methods

All samples were prepared by manually painting the geopolymer suspensions onto the underlying substrates using a brush. The microstructure of the coatings was observed on an Olympus SZ61 optical microscope and then on a 3D laser confocal microscope LEXT OLS5000 SAF (CLSM) and on a Vega 3 scanning electron microscope (SEM) from Tescan company (Tescan, Vega 3, Brno, Czech Republic). The thickness of the geopolymer liners was measured with a DeFelsko PosiTector 6000 portable coating thickness meter for metal substrates with an FNS type probe, with a measurement range of 0–1500 μm accuracy ± (1 μm + 1%) for a coating thickness of 0–50 μm, according to ISO 2360. The adhesion of the geopolymer to the chosen substrates was analyzed by the grid test method, according to ISO 2409 [36]—specifically, with the Elcometer 1542 grid test set.

## 3. Results and Discussion

### 3.1. Microstructure Analysis of Geopolymer Coatings Using SEM and CLSM

3.1.1. Geopolymer I Al

Figure 4 shows:

- Sample I 1L Al: The surface was slightly rough and showed microporosity. A small number of cracks are visible.
- Sample I 3L Al: The surface was also slightly rough. On this surface, there is also visible microporosity, but in a smaller amount compared to the previous sample, and cracks emerge from the porosity, which are larger than those in the sample I 1L Al. However, there were visible cracks that were stable, and no flaking was present.

- Sample I 5L Al: The surface was very rough and contained large porosity that looked like bubbles.
- Sample I S 5L Al: The surface was slightly rough. It contained porosity that looked like bubbles, like the previous sample, but in a small amount. The cracks were smaller compared to those of sample I 3L Al and stable.
- Sample I S 5L Al: The surface was slightly rough, and it was different compared to that of the other sample coatings of this series. The surface was granular, without any porosity or cracks. We assume that this structure is not ideal, and from previous research, we can predict weak adhesion behavior.

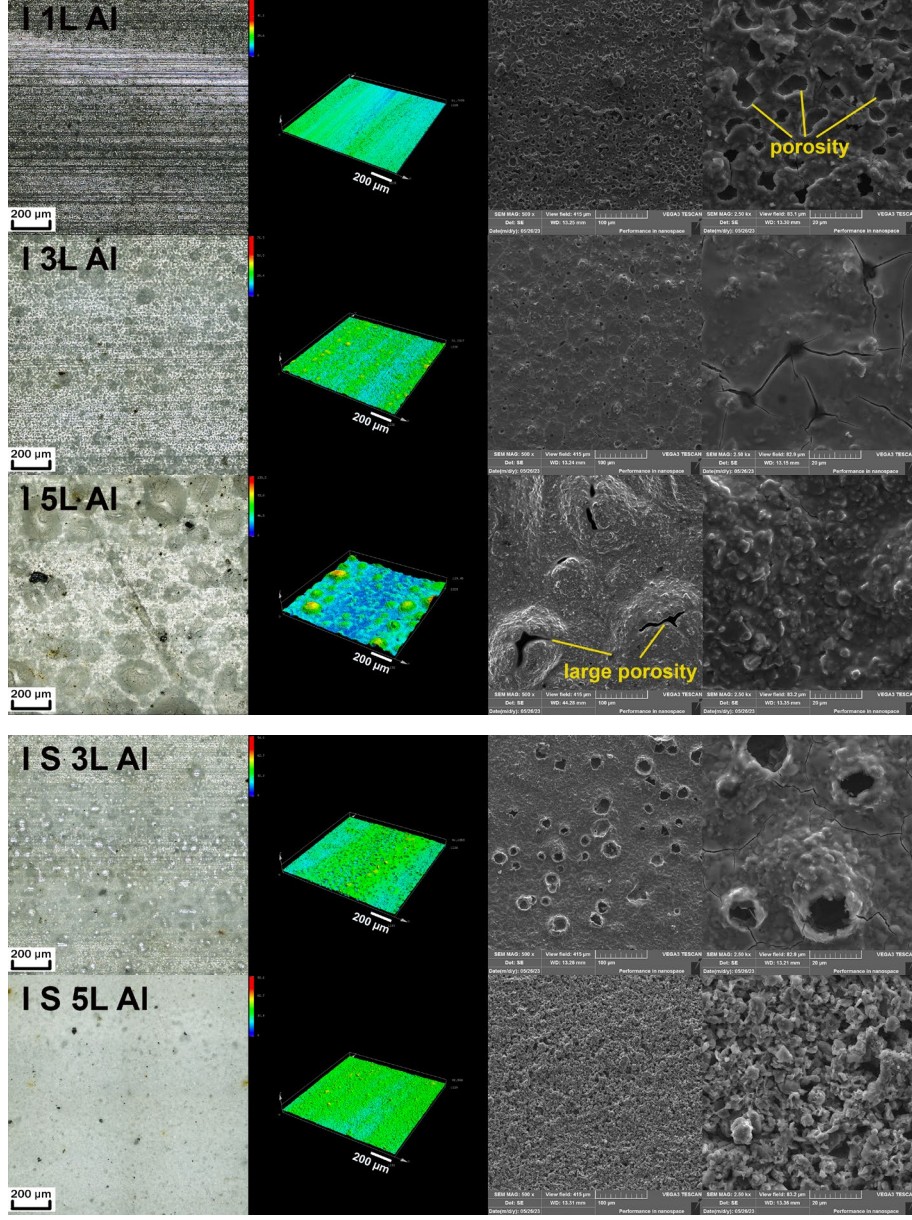

**Figure 4.** SEM and CLSM analysis of the surface of I geopolymer suspensions on the aluminum substrate.

Figure 4 shows the disparity in the surface structure between all of the analyzed samples. Even if they are coatings created from the same type of geopolymer suspension, which differ only in the number of created layers or in the curing method, the resulting surfaces are completely different. All coatings except for I S 5L Al exhibit some form of porosity. The I S 5L Al coating was granular and did not show any form of porosity. All

coatings are stable, the surface shows no flaking and the cracks are minimal and distributed homogeneously over the entire surface.

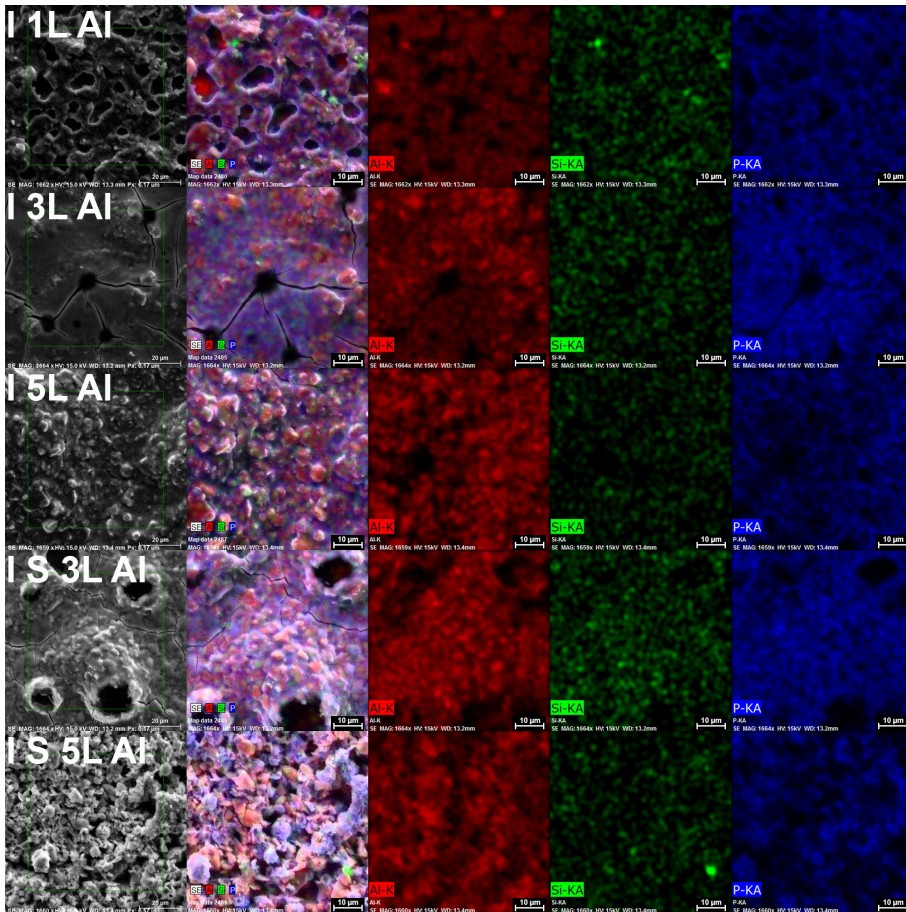

**Figure 5.** EDS mapping analysis of the surface of I geopolymer suspensions on the aluminum substrate.

The Figure 5 showed that except for the porosity positions, where, naturally, the presence of the elements analyzed using the EDS method was lower, it is evident that the elements Al, Si and P were distributed homogeneously, except for the sample I S 5L Al with a granular structure. It was observed that Al is more represented in the structure compared to Si, as the main elements of the geopolymer system.

3.1.2. Geopolymer I Fe

Figure 6 shows:

- Sample I 1L Fe: The coating surface of this sample is uniform, with a couple of small cracks. There is no visible porosity.
- Sample I 3L Fe: This surface is similar to sample I 1L Fe. It is evident that there is not much difference between a single-layer and multi-layer system, but we can observe that the cracks tend to coalesce to form long cracks.
- Sample I 5L Fe: This coating surface is the same as previous coatings on a steel substrate. However, it is completely without cracks.
- Sample I S 5L Fe: This sample is very different compared to the sample I 3L Fe. The surface is granular, with fine cracks.
- Sample I S 5L Fe: Even with this sample, the surface is different compared to that of the sample I 5L Fe, and a continuing trend can be seen with the layers applied in a different way (Figure 3). The layer is very fragmented and completely granular.

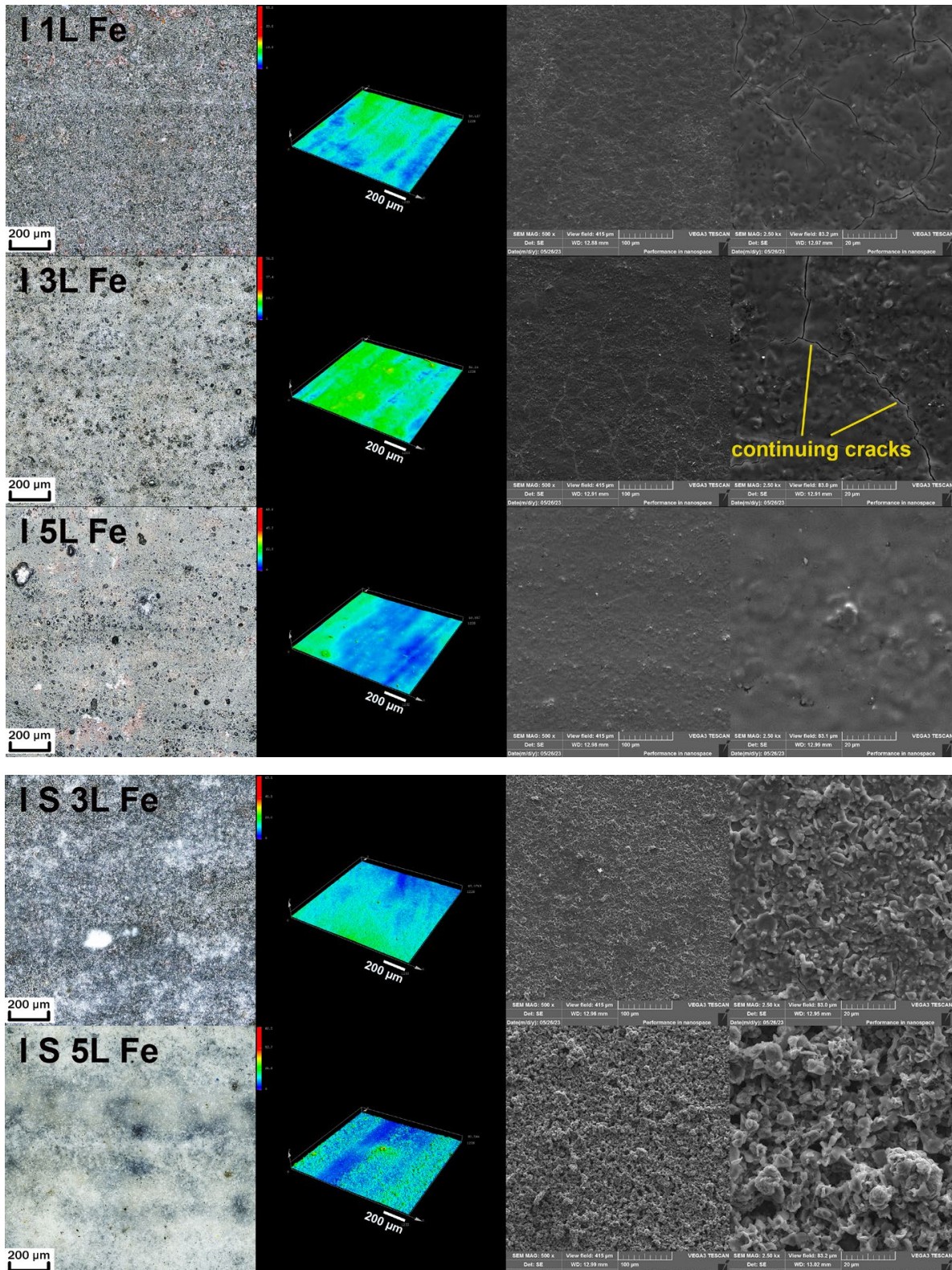

**Figure 6.** SEM and CLSM analysis of the surface of I geopolymer suspensions on the steel substrate.

In Figure 6, it was observed that no form of porosity occurs anymore in the coatings on the steel substrate. For the I S 5L Fe sample, a granular surface structure was again observed, just like the sample on the aluminum substrate in Figure 4. A lesser extent of the granular structure of the coating was also observed in the I S 5L Fe sample.

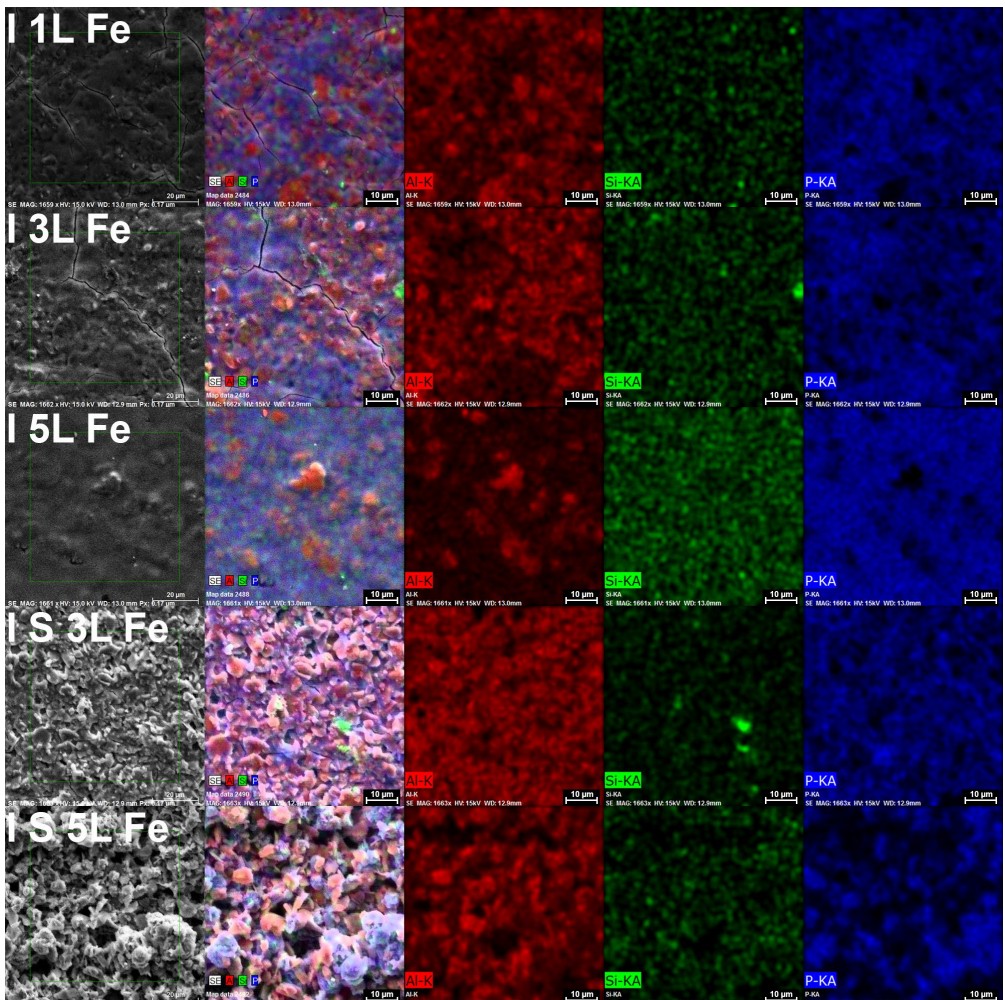

**Figure 7.** EDS mapping analysis of the surface of I geopolymer suspensions on the steel substrate.

Inhomogeneity was evident in all samples, even in samples without a granular structure. Figure 7 showed that the samples without a granular structure (I 1L Fe, I 3L Fe and I 5L Fe) have visible lumps on the surface, with an absence of P and a higher concentration of Al. The reason for this inhomogeneity was not studied, but it is probably a behavior of the geopolymerization and a property of the geopolymer. Application suspension by brush can also have an effect, since this phenomenon is also observed in the same suspension on the Al substrate, as described above. Si is distributed homogeneously.

### 3.1.3. Geopolymer J Al

Figure 8 shows the following:

- Sample J 1L Al: The surface had a very fine and smooth surface structure. It was stable, without visible cracks or flaking. There were visible lines after painting with a brush.
- Sample J 3L Al: There were also visible lines after painting with a brush; however, there were cracks on places with the thickest coating, which represents brighter places on the SEM picture.
- Sample J 5L Al: This surface was not as smooth as previous surfaces. There were visibly larger cracks than in the previous sample, which were visible mainly in places with a thicker layer of coating, but no flaking.
- Sample J S 3L Al: This surface was also not smooth. Cracks were visible.
- Sample J S 5L Al: The surface was slightly rough. On this surface, there were visible, large cracks. Here, there was also visible flaking for the first time.

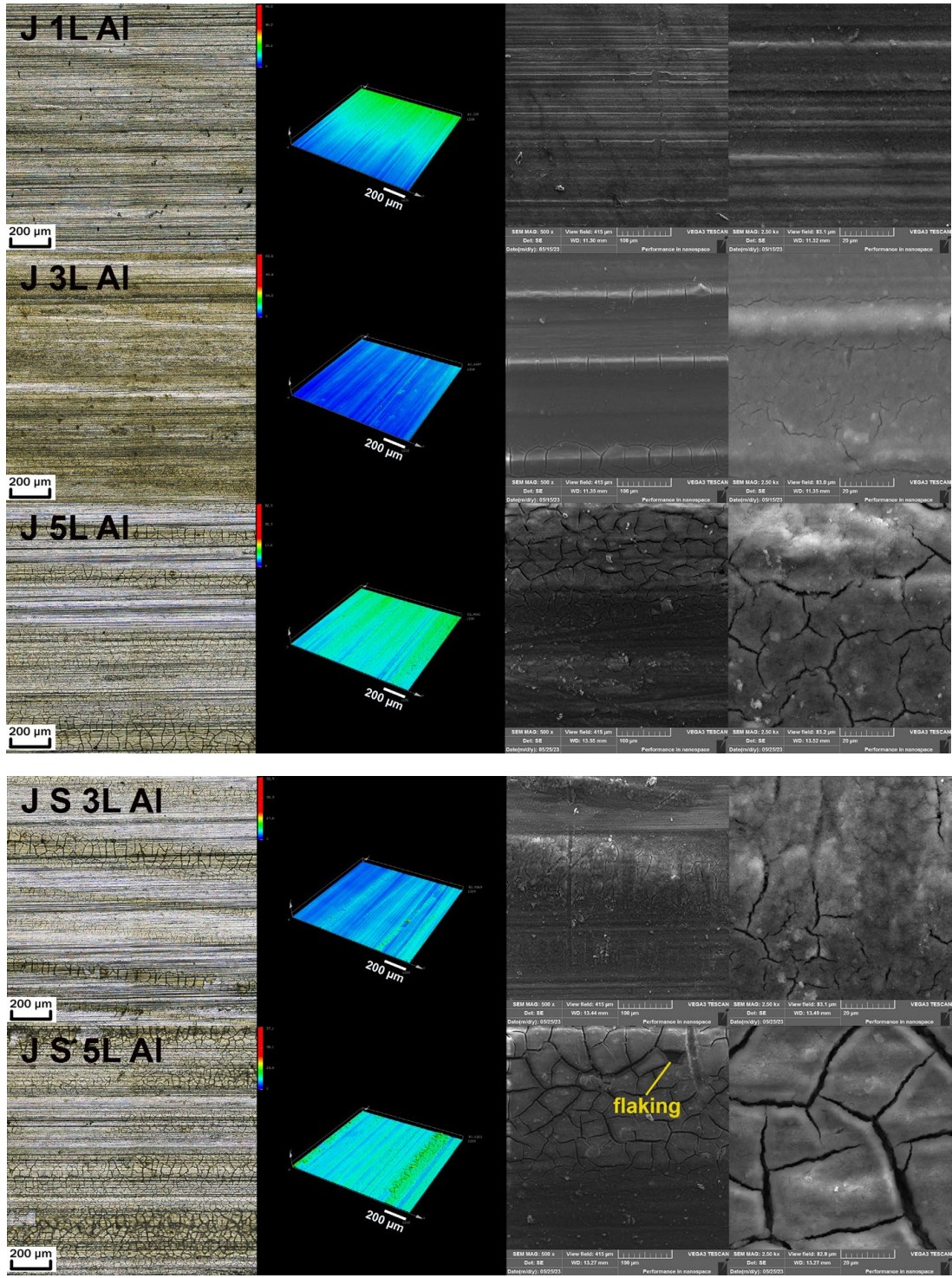

**Figure 8.** SEM and CLSM analysis of the surface of J geopolymer suspensions on the aluminum substrate.

The GP J coatings on the aluminum substrate show different surface structures compared to the GP I coatings in Figure 4. In the GP J coatings, there were not any signs of porosity or granular structures. The J S 5L Al sample showed a considerable number of cracks. The same J 5L Al multi-layer sample, where the layers were cured sequentially, showed fewer cracks, and spalling was present only in the sample J S 5L Al.

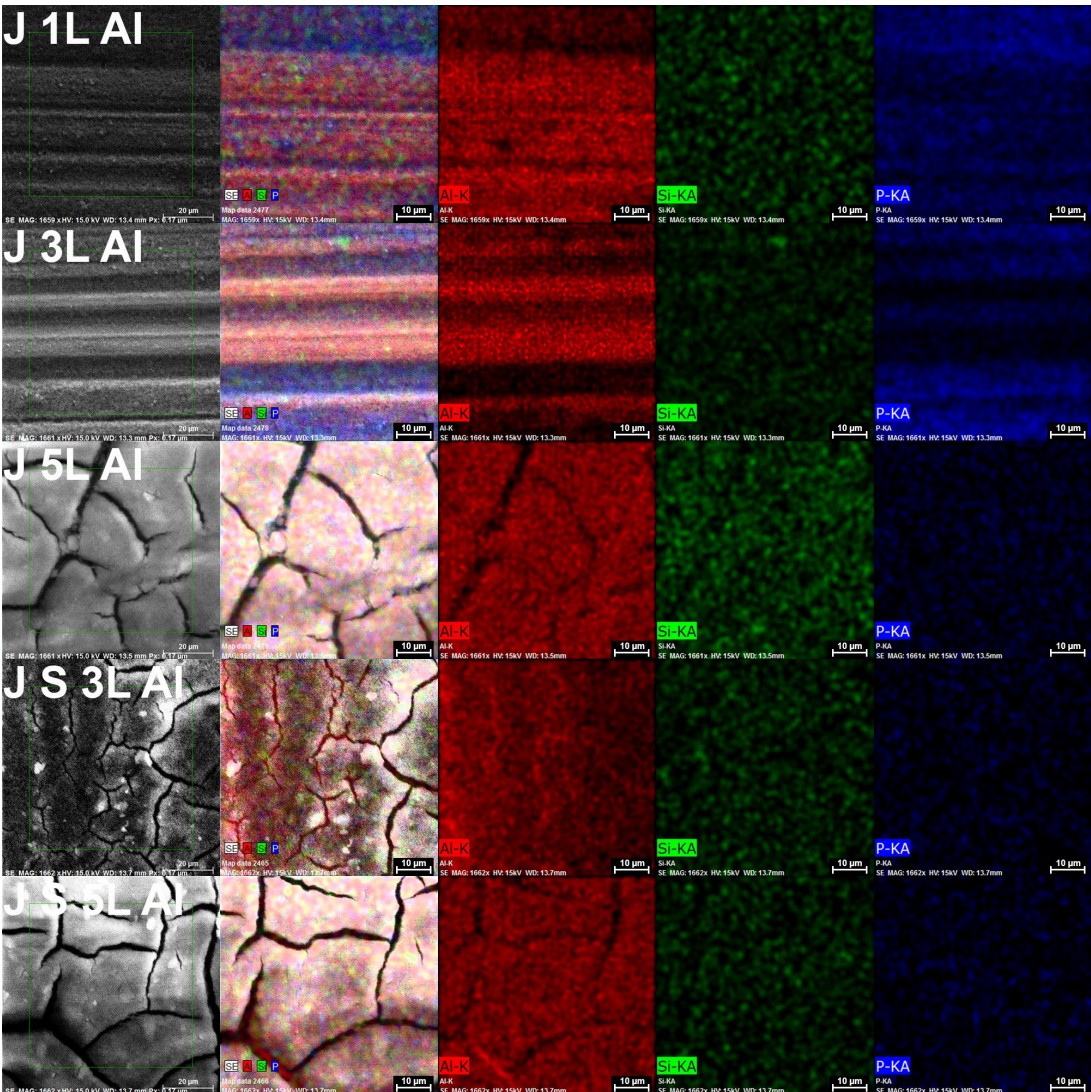

**Figure 9.** EDS mapping analysis of the surface of J geopolymer suspensions on the aluminum substrate.

The inhomogeneity of the distribution of Al and P elements which could be seen on Figure 9 was clearly visible on the J 1L Al and J 3L Al samples in the lines, which were created after the application of the geopolymer by brush. These lines were characteristic of a thinner layer of the geopolymer coating. The lines were enriched more with phosphorous, and a smaller presence of aluminum was evident.

### 3.1.4. Geopolymer J Fe

Figure 10 shows:

- Sample J 1L Fe: The surface had a very fine and smooth surface structure. Cracks were regular and visible only in the detail.
- Sample J 3L Fe: The surface was also very fine, with a smooth structure. Cracks were larger but without flaking.
- Sample J 5L Fe: The cracks of this multi-layer system were larger, without flaking.
- Sample J S 3L Fe: The cracks were extensive. No flaking was observed, but the space of the cracks was large and seemed unstable. More cracks were situated in the places with a thicker coating (after the brush application).
- Sample J S 5L Fe: The surface was rough. The cracks were extensive and homogeneously distributed over the surface, not only in places with a thicker coating.

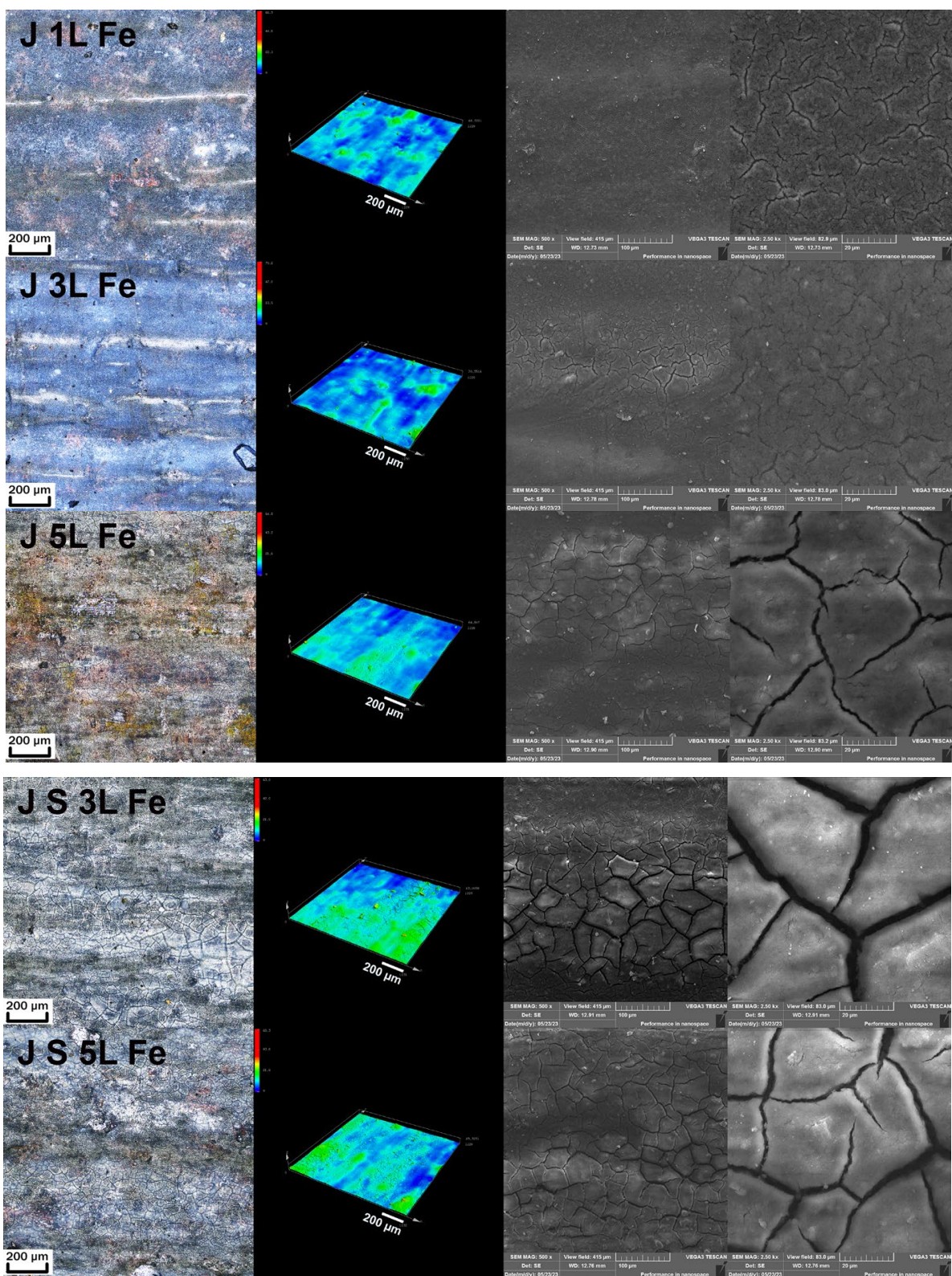

**Figure 10.** SEM and CLSM analysis of the surface of J geopolymer suspensions on the steel substrate.

Here, in Figure 10, cracks were observed in all of the measured samples. Their intensity increased gradually with the increasing number of layers. No porosity or granular structure of the coating was visible. The surfaces for all samples are very similar to the surfaces on the aluminum substrate.

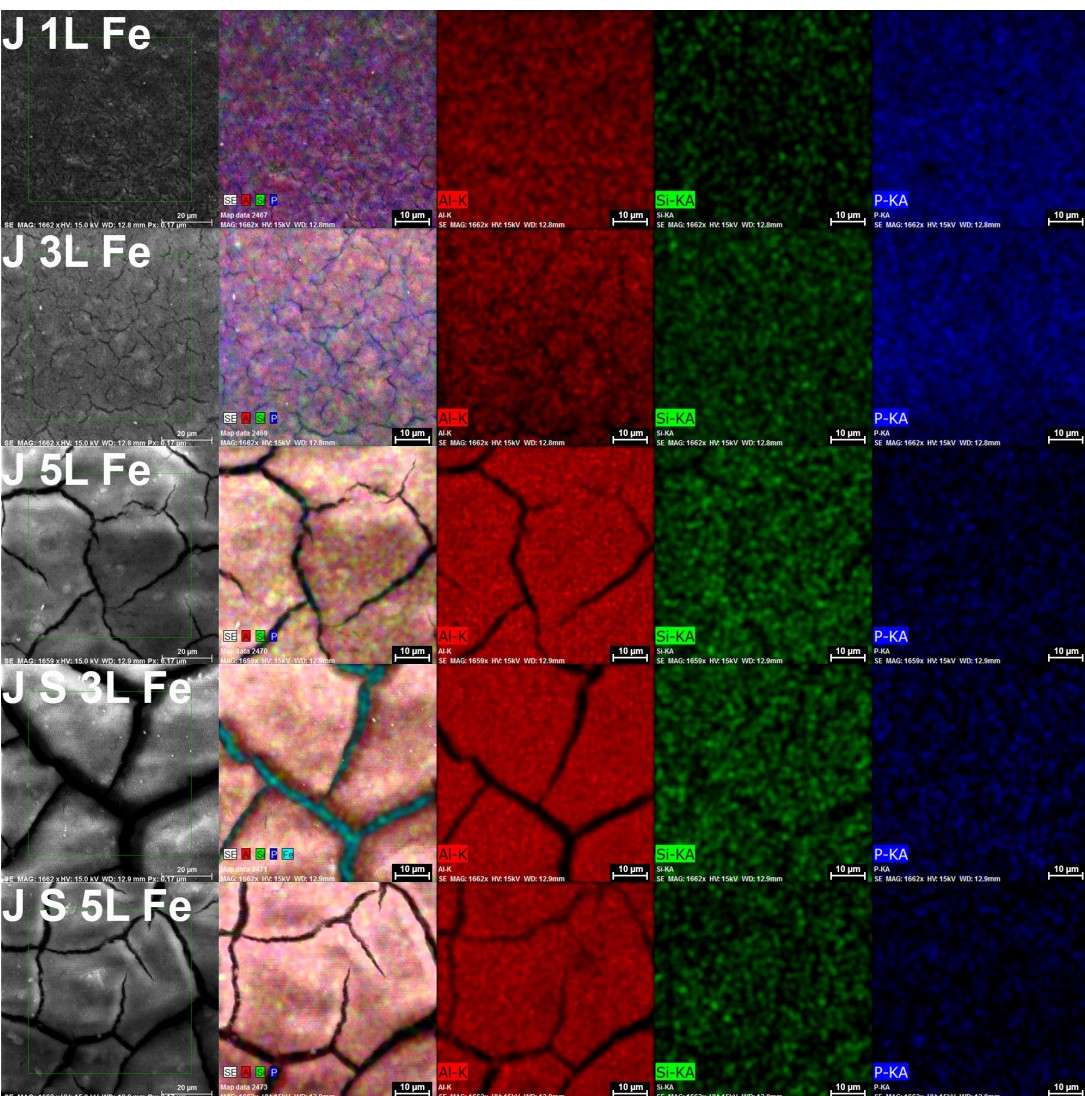

**Figure 11.** EDS mapping analysis of the surface of J geopolymer suspensions on the steel substrate.

EDS analysis seen on Figure 11 showed a homogeneous distribution of all elements. Homogeneity was the most pronounced of all the samples examined. No representation of elements was visible in the area of the cracks. The sample J S 3L Fe showed that, through the cracks, the steel substrate was visible and was in contact with atmospheric conditions. These coatings cannot provide full chemical protection of the substrate. This sample was chosen to present this phenomenon. The beam intensity of the SEM microscope could not penetrate the entire coating [37].

### 3.2. Analysis of the Adhesion of the Geopolymer Layer by the Grid Test

The achieved results of the grid test are shown in Figure 12, and the evaluation of the results is shown in Table 2. Table 2 shows the evaluated results of the grid test of various GP on Al and Fe substrates. GP I and J on the Al substrate achieved a rating of one, but only GP I S 5L had a rating of two, most likely due to too large of a layer of the GP suspension in combination with the method of application and curing. GP I on the Fe substrate achieved a worse rating of three, due to the presence of a natural oxide layer on the surface of the substrate and its peeling, together with the coating. GP J on the Al substrate achieved a rating of one for all samples and layers. GP J on the Fe substrate showed a better value of the grid test, with a rating of one for J 5L and a rating of two for J S 3L. The other GP J had a rating of three.

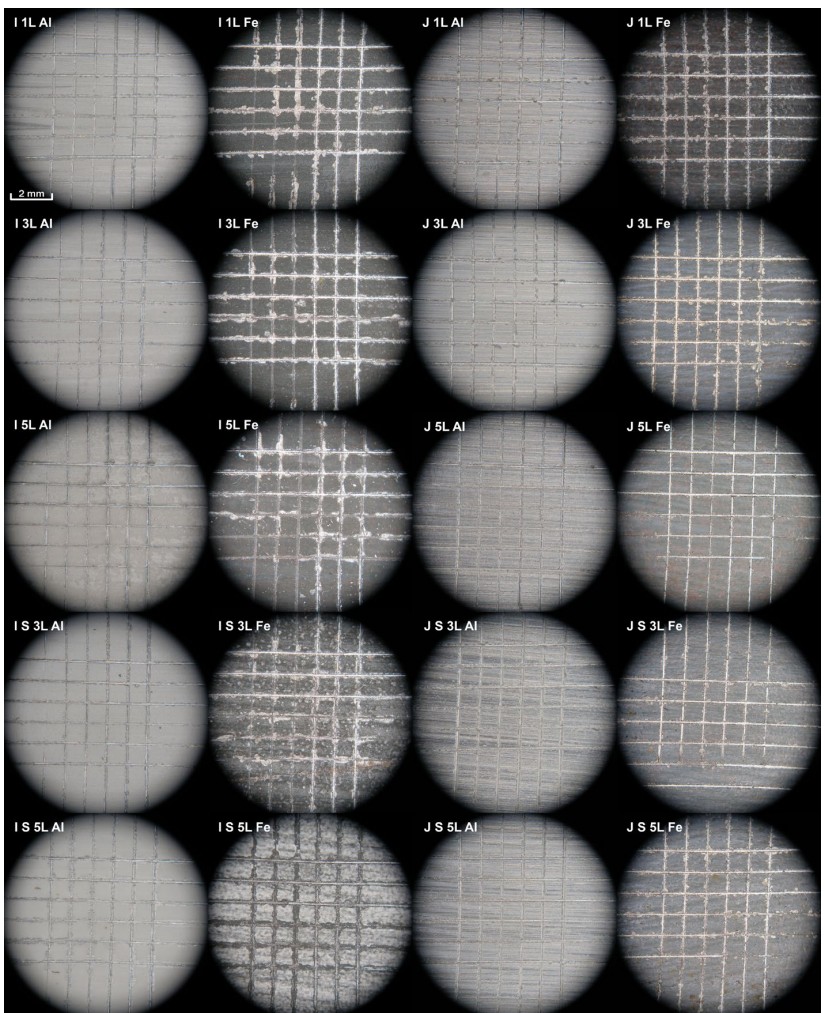

**Figure 12.** Grid test of GP coatings on the Al and Fe substrates.

**Table 2.** Grid test rates of GP coatings on the Al and Fe substrates.

| Substrate | I 1L | I 3L | I 5L | I S 5L | I S 5L | J 1L | J 3L | J 5L | J S 3L | J S 5L |
|---|---|---|---|---|---|---|---|---|---|---|
| Al | 1 | 1 | 1 | 1 | 2 | 1 | 1 | 1 | 1 | 1 |
| Fe | 3 | 3 | 3 | 3 | 3 | 3 | 3 | 1 | 2 | 3 |

The results of the GP I 1L Al and GP J 1L Al + Fe coatings corresponded with our previous research [23,24], but GP I 1L Fe had worse ratings than those in previous research. Due to the fact that no pre-treatment of the surface before the application of GP suspensions took place on the substrates (except for surface degreasing), there are natural oxides on both types of substrates. As can be seen from the grid test, the natural $Al_2O_3$ oxide layer on the Al substrate does not have a negative effect on the adhesion of the GP coatings to the surface. The Fe substrate is also covered with a natural oxide layer that is thicker than that on the Al substrate. According to the XRD analysis, it was found that $Fe_3O_4$, $Fe_2O_3$ and FeO oxides are found on the surface of the Fe substrate. The lower adhesion of GP coatings on the steel substrate is apparently caused by this oxide layer, which does not have high adhesion to the substrate itself (the steel layer located below this layer) and thus peels off from the surface together with the GP coating.

### 3.3. Analysis of the Thickness of the Geopolymer Layer

A comparison of the measured values of the thicknesses of all layers for GP I on the aluminum and steel substrates can be seen in Figure 13. Single-layer coatings reach

almost the same thickness for both types of substrates (11% difference). The resulting thickness of I 1L Al is approximately 2.7 times higher than the thickness achieved in previous research [23]. All samples on both types of substrates show an almost linear increase in the layer thickness with an increasing number of layers. It can be seen from the graph that overall thicker layers were achieved on the aluminum substrate than on the steel substrate (about 20%–30%). What is interesting is the comparison of the multi-layer sample I 3L with I S 5L and that of I 5L with I S 5L, where a clear trend can be observed, where the samples cured after each layer reach an overall lower thickness than the samples cured after the last-applied layer.

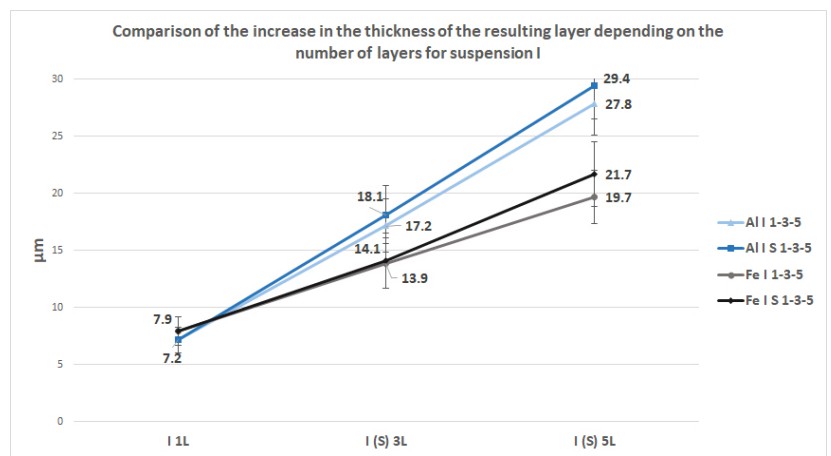

**Figure 13.** Comparison of the increase in the thickness of the resulting layer depending on the number of layers for suspension I.

The thicknesses of the GP J layers on both types of substrates are shown in Figure 14. As we can see, the J 1L Al single-layer sample reaches a thickness of 3.4 µm, which, as in the rare case of GP I, is an increase in thickness compared to previous research [23] by 2.3 times. Sample J 1L Fe has a greater thickness for this geopolymer, almost twice that of the aluminum substrate. All the layers on the Fe substrate reach greater thicknesses by about 70%–110% compared to the Al substrate, which is the opposite trend to that of GP I. The comparison of the multi-layer sample J 3L with J S 3L and that of J 5L with J S 5L again confirm the previous trend, where the samples cured after each layer reach a lower thickness than the samples cured after the last layer (except for J 3L and J S 3L on the Al substrate, where the thickness is almost identical).

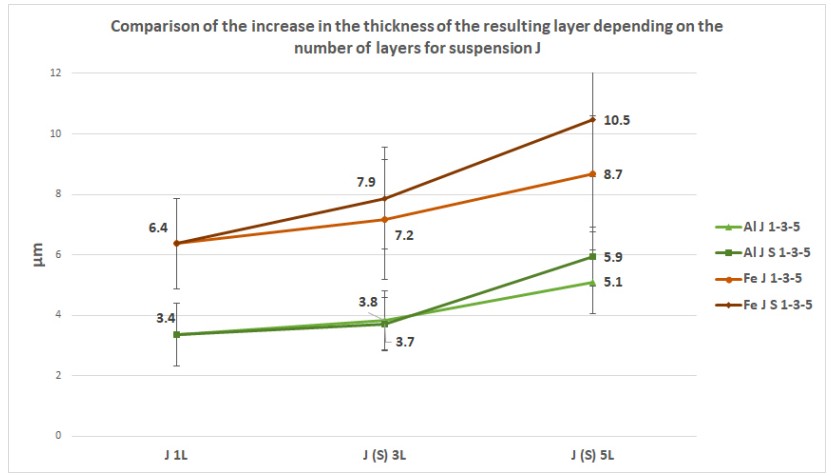

**Figure 14.** Comparison of the increase in the thickness of the resulting layer depending on the number of layers for suspension J.

## 4. Conclusions

The basis of this research was the creation of geopolymer coatings on aluminum and construction steel substrates, which were applied to the surface using a brush. A thicker layer of the suspension was already created on the substrates during the application itself, which was supposed to verify the properties and surface of the thicker layers created in this way, which follows previous research that, on the contrary, was focused on creating very thin layers [23,24]. Two types of geopolymer suspensions, I and J, were selected from the previous research. Furthermore, GP suspensions were applied in multiple layers, three and five, and two different methods of curing were used (see Section 2.2).

The formed coatings showed a certain porosity, and, above all, they are prone to the formation of cracks. These cracks are created naturally during curing by the emission of water from the volume and the different thermal expansion of the geopolymeric suspension and the underlying substrate. Cracks generally increased with the increasing number of layers. However, in most cases, the cracks did not appear to affect the cohesion of the coating or the adhesion of the coatings to the substrate surface. Thus, the porosity, rough surface and cracks do not have to reduce the resulting properties or the use of the coating, and in some cases, they can be beneficial. A rough or cracked surface can show better adhesion, e.g., when using glue in glued joints, when a larger surface is needed for the good adhesion of the joint [37]. A rough surface can help improve part handling and increase safety [38]. A jagged, rough or cracked surface change the optical properties of the surface so that, for example, there are no reflections of light from the surface [39]. Self-lubricating systems appear to be a very suitable application for this type of surface, where the surface of the part is provided with this coating, which contains many cracks and capillaries into which the lubricant is applied, and then gets between the functional surfaces and thus affects the tribological properties, e.g., by reducing friction, which leads to an increase in the life of the component and a reduction in the need for maintenance [40–42].

The application of suspensions with a brush is economical, but according to microscopic analysis, it is evident that it introduces a certain inhomogeneity into the coating, whether it is the fluctuating thickness of the layer, which then causes cracking, or the inhomogeneity in the distribution of Al, Si and P elements in the coating.

The grid test confirmed the very high adhesion of GP coatings on the aluminum substrate, independent of the thickness, the number of layers and the method of curing. On the steel substrate, the adhesion of the coatings was lower and partly dependent on the above-mentioned variables.

Observing the thickness of the layer on the underlying substrate is an important aspect. Thicker layers protect the underlying substrate better against corrosion and mechanical wear. As can be seen from the electron microscopy images, cracks on the surface increased in size with the increasing number of layers. Cracks of the underlying substrate are undesirable in the case of the application of GP anti-corrosion suspensions, as they reduce corrosion resistance (the corrosive environment with cracks can reach the underlying substrate). Large cracks are undesirable because they lead to the peeling of the GP layer from the underlying substrate, which was observed in some cases.

Finding a balance between the layer thickness and cracks is essential for future applications.

By applying a thicker layer of the GP suspension when applied with a brush, a final, thicker layer could be created, which, according to analyses, had no negative effect on the quality of the resulting surface. By applying additional layers, thicker layers could be created, which could affect other properties (mechanical, chemical) of the resulting surface. The joint thickness is further influenced by both the composition of the geopolymer suspension and the method of the application and curing of single layers in multi-layer samples.

To increase the homogeneity of the distribution of individual elements in the resulting layers and further improve the quality of the surface, in terms of, e.g., reducing the roughness or reducing the heterogeneity of the layer thickness in different places of the surface, which is caused by an application with a brush, and, thus, to reduce the formation, number and size of cracks and fissures, which are apparently caused by too thick of an

applied suspension (again, in certain places), it would be necessary to change the method of application of the suspension. As a suitable solution to these inhomogeneities during the application, the suspension could be applied by spraying (air brush) or possibly even by means of a roller. These application methods (mainly the air brush) are more complicated and complex. Other forms of applications, e.g., dipping, are not suitable due to the need to maintain a very small thickness of the applied suspension. Moreover, the search for another application method contradicts the basic assumption of cheap and simple painting.

This research shows that it is possible to create multi-layer geopolymer coatings that achieve a good surface quality and adhesion on various underlying substrates while maintaining a relatively small thickness. This can be advantageous, for example, for functional components, where after the application of several layers of GP suspensions, there will be no large dimensional change, and the possible functionality is thus not affected. To further increase the quality of the surface, it would be advisable, for example, to change the method of application of the suspension in order to achieve a more even coverage of the surface of the substrate, which will ultimately affect the homogeneity of the resulting layers. The lower adhesion of the layers on the steel substrate could probably be solved by a suitable mechanical (or even chemical) pre-treatment of the surface, when the surface oxide layer is removed. Adhesion is excellent with the aluminum substrate, and the oxide layer does not seem to negatively affect adhesion. Furthermore, it is possible to focus on the final temperature when curing the layers. In this research, it was applied at a curing temperature of 170 °C, but it is possible to change this temperature further, which can affect the resulting mechanical properties of the applied layers. Lowering the resulting temperature and shortening the holding time will have a positive effect on production costs, but at the same time, the temperature must not be too low in order for the applied layers to properly geopolymerize. On the contrary, a higher temperature can further positively affect the mechanical properties of the layers, e.g., by increasing the surface hardness (mainly in the case of GP suspensions with $Al_2O_3$ content). Continuing that research and the previous research [23,24], the focus may be on the analysis of the mechanical properties of multilayer coatings, such as the microhardness of the surface of the layers or the tribological properties [40,42] (possibly the corrosion resistance of the coatings [43]).

**Author Contributions:** Conceptualization, M.J. and F.M.; Data curation, M.J. and J.N.; Formal analysis, F.M., J.M. and A.M.; Funding acquisition, M.J. and J.N.; Investigation, M.J., F.M. and J.M.; Methodology, M.J., F.M. and J.M.; Project administration, M.J. and F.M.; Resources, M.J., F.M. and J.M.; Supervision, J.N.; Validation, M.J. and J.N.; Visualization, M.J., F.M. and J.M.; Writing—original draft, M.J., F.M. and J.M.; Writing—review and editing, M.J., F.M., J.M. and A.M. All authors have read and agreed to the published version of the manuscript.

**Funding:** This research was funded by NANOTECH ITI II. No. CZ.02.1.01/0.0/0.0/18_069/0010045 and the internal UJEP Grant Agency (UJEP-SGS-2022-48-002-2).

**Data Availability Statement:** Data sharing is not applicable to this article.

**Conflicts of Interest:** The authors declare no conflict of interest.

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
