# Peer review of "Thickness, Adhesion and Microscopic Analysis of the Surface Structure of Single-Layer and Multi-Layer Metakaolin-Based Geopolymer Coatings"

_coatings, doi:10.3390/coatings13101731_

Round 1

Reviewer 1 Report

The manuscript reports on the surface structure of single- and multi-layer metakaolin based geopolymer coatings. The authors used two types of dispersions to create coatings, namely metakaolin with AlOH3 or Al2O3 powder in mixture of H3PO4 and isopropyl alcohol solution. Coatings were applied by a simple method using a brush. The article discusses the properties of the obtained geopolymer coatings depending on the number of layers and the method of their curing. The undoubted advantage of the study is the EDS mapping analysis of the surfaces.

Despite the interesting results reported in the manuscript, in my opinion, it should be significantly improved. Therefore, I recommend to the authors revise the manuscript according to some issues listed below.

1) First of all, I recommend paying more attention to the discussion of the obtained results and the observed features. For example, the mechanism of crack formation and their absence in other cases. It should be summed up, which is a good result. How many layers should be applied by which method and for what purposes? Otherwise, it seems that the only conclusion in the manuscript is that the brush application technique is bad in itself and other methods should be tried, such as airbrush etc., lines 562-572. In this vein, it is not clear why the study was carried out at all.

2) I am slightly confused by the interpretation of the results, lines 273-275 and 311-314. What does insufficient dissolution of metakaolin mean? Were the solutions for each method prepared at different times? Is insufficient dissolution typical for all solutions or only for particular one? In this case, the sample is not relevant and cannot be compared with the rest of the series. Moreover, if the quality and composition of the coating solution is not controlled, what is the subject of the manuscript? The results as presented do not show a correlation between coating morphology and application methods.

3) Line 400, the conclusion does not show the correlation with the experimental results. For iron substrates, almost all samples exhibit low adhesion, regardless of the granularity of the structure.

4) The parameters "R" in Table 2 must be deciphered.

5) The scale bar in Figures 4-11 is unreadable.

6) Figures 13 and 14, as well as 15 and 16, duplicate data. There is no error bar in Figures 14 and 16.

7) Line 449, the authors note an increasing trend. It is incorrect to draw such a conclusion based on 3 samples.

Some minor comments:

8) Table 1 is redundant. The designation of geopolymer suspensions is perfectly clear from the text. The same goes for Figure 3.

9) Figure 1 and partially the data of Table 2 were previously published in ref. [27]. I comprehend, that the authors used the same substrate, however, another similar image should be provided.

10) Column "units" in Table 2 is duplicated.

11) Some typos should be fixed. For example, captions for Figures 4-6, lines 237, 267, 284. The Figures are for I geopolymer, but not for J.

12) The abbreviation MFT should be deciphered, line 103.

13) The description of iPrOH (isopropyl alcohol) should be given at the first mention in the text in line 120, but not in line 126.

14) For future research. How will the surface morphology change with vertical coating? It would be interesting to study the rheological behavior of suspensions, which is especially important for practical applications.

Minor editing of English language required.

Author Response

Changes in the manuscript are highlighted in green.

1) First of all, I recommend paying more attention to the discussion of the obtained results and the observed features. For example, the mechanism of crack formation and their absence in other cases. It should be summed up, which is a good result. How many layers should be applied by which method and for what purposes? Otherwise, it seems that the only conclusion in the manuscript is that the brush application technique is bad in itself and other methods should be tried, such as airbrush etc., lines 562-572. In this vein, it is not clear why the study was carried out at all.

  • A discussion about crack formation has been added to the conclusion.

2) I am slightly confused by the interpretation of the results, lines 273-275 and 311-314. What does insufficient dissolution of metakaolin mean? Were the solutions for each method prepared at different times? Is insufficient dissolution typical for all solutions or only for particular one? In this case, the sample is not relevant and cannot be compared with the rest of the series. Moreover, if the quality and composition of the coating solution is not controlled, what is the subject of the manuscript? The results as presented do not show a correlation between coating morphology and application methods.

  • Note: insufficient dissolution of metakaolin, has been re-evaluated and rewritten.

3) Line 400, the conclusion does not show the correlation with the experimental results. For iron substrates, almost all samples exhibit low adhesion, regardless of the granularity of the structure.

  • Added justification of poor GP adhesion with Fe substrate and I S 5L Al sample.

4) The parameters "R" in Table 2 must be deciphered.

  • The parameters have been described.

5) The scale bar in Figures 4-11 is unreadable.

  • Scales have been adjusted.

6) Figures 13 and 14, as well as 15 and 16, duplicate data. There is no error bar in Figures 14 and 16.

  • Images have been corrected.

7) Line 449, the authors note an increasing trend. It is incorrect to draw such a conclusion based on 3 samples.

  • Edited, removed.

Some minor comments:

8) Table 1 is redundant. The designation of geopolymer suspensions is perfectly clear from the text. The same goes for Figure 3.

  • The table is redundant and has been removed.

9) Figure 1 and partially the data of Table 2 were previously published in ref. [27]. I comprehend, that the authors used the same substrate, however, another similar image should be provided.

  • Added citation to original source.

10) Column "units" in Table 2 is duplicated.

  • The column has been fixed.

11) Some typos should be fixed. For example, captions for Figures 4-6, lines 237, 267, 284. The Figures are for I geopolymer, but not for J.

  • Images have been corrected.

12) The abbreviation MFT should be deciphered, line 103.

  • The shortcut has been entered.

13) The description of iPrOH (isopropyl alcohol) should be given at the first mention in the text in line 120, but not in line 126.

  • The shortcut has been entered.

14) For future research. How will the surface morphology change with vertical coating? It would be interesting to study the rheological behavior of suspensions, which is especially important for practical applications.

  • This is an interesting insight for future research. Practical application is a key point of this study.

Reviewer 2 Report

The authors presented an article about “Thickness, adhesion and microscopic analysis of surface structure of single-layer and multi-layer metakaolin based geopolymer coatings.”

I believe that the authors have done a successful study on coating. However, I think that the novelty of the study remains in the background because its difference from a similar study published in the same journal has not been better demonstrated. The article appeared to be written carefully in general terms. In general, although the text parts in the figures could not be read, it was seen to be in good order.. I also think it will be a source of work for future publications. I think the paper is well organized and appropriate for the “Coatings” journal, but the paper will be ready for publication after minor revision.

ü  The abstract looks good. Please include all significant numerical results.

ü  What is the problem? Why was the manuscript written? Please explain the reason in the introduction part. In the last paragraph of the introduction, the novelty of the study and the differences from the past in detail should be expressed.

ü  In the study, no literature study was mentioned for the use of H3PO4 acid with Al(OH)3 and H3PO4 acid with nano Al2O3. Are these materials used for the first time? The literature study on the use of similar materials should be expanded.

ü  Please provide a table containing the physical and chemical properties for the Al2O3 nano powder used in the materials and methods section.

ü  The image of the Al material in Figure 1 should be cited as it was used in a previous article.

ü  Using too many citations in the material method section may distract the reader. This leads to a decrease in the quality of the article. Please avoid using unnecessary citations.

ü  Results scales are not read in SEM, CLSM, and EDS result figures. Please increase their visibility and figure quality.

ü  The length of the conclusion is unacceptable. Please give the results of the study in general terms.

ü  How was the temperature parameter determined? Please specify in the article.

ü  The paper is well-organized, yet there is a reference problem. Cited sources should be primary ones. Namely, the indexed area shows the power of a paper and directly your paper’s reliability. Please make regulations in this direction.

ü  Please fix the typographical and eventual language problems in the paper.

*** Authors must consider them properly before submitting the revised manuscript. A point-by-point reply is required when the revised files are submitted.

ü  Please fix the typographical and eventual language problems in the paper.

Author Response

Changes in the manuscript are highlighted in blue.

ü  The abstract looks good. Please include all significant numerical results.

  • Abstract has been improved.

ü  What is the problem? Why was the manuscript written? Please explain the reason in the introduction part. In the last paragraph of the introduction, the novelty of the study and the differences from the past in detail should be expressed.

  • This section in the introduction has been edited and developer.

ü  In the study, no literature study was mentioned for the use of H3PO4 acid with Al(OH)3 and H3PO4 acid with nano Al2O3. Are these materials used for the first time? The literature study on the use of similar materials should be expanded.

  • The use of H3PO4 acid with Al(OH)3 and Al2O3 has an innovative character (Al(OH)3 is a fire retarder) and Al2O3 shows high hardness. Text explaining its use has been added.

ü  Please provide a table containing the physical and chemical properties for the Al2O3 nano powder used in the materials and methods section.

  • All geopolymer suspensions were delivered by IIC CAS in Rez without detailed information about it. We can add XRD and XRF analysis of solid ingredients.

ü  The image of the Al material in Figure 1 should be cited as it was used in a previous article.

  • Citation has been added.

ü  Using too many citations in the material method section may distract the reader. This leads to a decrease in the quality of the article. Please avoid using unnecessary citations.

  • Citations has been revised.

ü  Results scales are not read in SEM, CLSM, and EDS result figures. Please increase their visibility and figure quality.

  • Scales have been adjusted.

ü  The length of the conclusion is unacceptable. Please give the results of the study in general terms.

  • The conclusion was shortened and reworked.

ü  How was the temperature parameter determined? Please specify in the article.

  • The temperature parameter was determined experimentally in the previous research according to citation [23], and this temperature was maintained due to direct continuity and previous research.

ü  The paper is well-organized, yet there is a reference problem. Cited sources should be primary ones. Namely, the indexed area shows the power of a paper and directly your paper’s reliability. Please make regulations in this direction.

  • Citations has been revised.

ü  Please fix the typographical and eventual language problems in the paper.

  • Language corrections have been made.

Reviewer 3 Report

Manuscript ID: coatings-2604235

Title: Thickness, adhesion and microscopic analysis of surface structure of

single-layer and multi-layer metakaolin based geopolymer coatings 

Authors: Martin Jaskevic *, Jan Novotny, Filip Mamon, Jakub Mares, Angelos

Markopoulos

The manuscript explores the morphological properties of metakaolin-based geopolymer coatings on the naturally oxidized surfaces of aluminum alloy EN-AW 6060 and structural steel 1.0038. H3PO4 acid was used to activate the polymerization process. The manuscript is easy to read. The main shortcoming of the manuscript is the lack of justification for the purpose of the study. Which alloy surface property should be improved by the new coating?

Minor changes required.

Why were aluminum and steel surfaces chosen as model systems? The introduction does not answer this question.

What are Ri (i = a, z, max, t) in Table 2?

Authors should re-read the text and combine sections of the text that discuss the same topic but are fragmented throughout the text of the manuscript. See, the coating method (91-97, 158-161), the composition of the geopolymer suspensions (119-127).

Figure 4. Change J to I.

The final part is a repetition of all the details of the results obtained in the study. Instead, the most important overall results should be summarized.

Author Response

Changes in the manuscript are highlighted in yellow.

Why were aluminum and steel surfaces chosen as model systems? The introduction does not answer this question.

  • We selected the most used Al and Fe materials, which we chose as the basic materials for the beginning of our research.

What are Ri (i = a, z, max, t) in Table 2?

  • The Ri (i = a, z, max, t) parameters have been described.

Authors should re-read the text and combine sections of the text that discuss the same topic but are fragmented throughout the text of the manuscript. See, the coating method (91-97, 158-161), the composition of the geopolymer suspensions (119-127).

  • These parts have been rewritten.

Figure 4. Change J to I.

  • Image description changed

The final part is a repetition of all the details of the results obtained in the study. Instead, the most important overall results should be summarized.

  • The conclusion was shortened and reworked.

Round 2

Reviewer 1 Report

The authors have responded to my comments and the manuscript can be accepted for publication.

Minor corrections are required.

Reviewer 3 Report

I have no more comments that would necessitate another review cycle.